# Biosafety of Genome Editing Applications in Plant Breeding: Considerations for a Focused Case-Specific Risk Assessment in the EU

**DOI:** 10.3390/biotech10030010

**Published:** 2021-06-22

**Authors:** Michael F. Eckerstorfer, Marcin Grabowski, Matteo Lener, Margret Engelhard, Samson Simon, Marion Dolezel, Andreas Heissenberger, Christoph Lüthi

**Affiliations:** 1Umweltbundesamt–Environment Agency Austria (EAA), Landuse & Biosafety Unit, Spittelauer Lände 5, 1090 Vienna, Austria; marion.dolezel@umweltbundesamt.at (M.D.); andreas.heissenberger@umweltbundesamt.at (A.H.); 2Ministry of Climate and Environment, Department Nature Conservation, GMO Unit, Wawelska 52/54, 00-922 Warszawa, Poland; marcin.grabowski@srodowisko.gov.pl; 3ISPRA (Italian Institute for Environmental Protection and Research), Department for Environmental Monitoring and Protection and for Biodiversity Conservation, Via Vitaliano Brancati, 48, 00144 Roma, Italy; matteo.lener@isprambiente.it; 4Federal Agency for Nature Conservation, Division of Assessment of GMOs/Enforcement of Genetic Engineering Act, Konstantinstr. 110, 53179 Bonn, Germany; Margret.Engelhard@BfN.de (M.E.); Samson.Simon@BfN.de (S.S.); 5Federal Office for the Environment (FOEN), Biotechnology Section, Soil and Biotechnology Division, BAFU, CH-3003 Bern, Switzerland; Christoph.Luethi@bafu.admin.ch

**Keywords:** novel genomic techniques, genome editing, CRISPR/Cas, plant modification, GMO, environmental risk assessment, biosafety regulation

## Abstract

An intensely debated question is whether or how a mandatory environmental risk assessment (ERA) should be conducted for plants obtained through novel genomic techniques, including genome editing (GE). Some countries have already exempted certain types of GE applications from their regulations addressing genetically modified organisms (GMOs). In the European Union, the European Court of Justice confirmed in 2018 that plants developed by novel genomic techniques for directed mutagenesis are regulated as GMOs. Thus, they have to undergo an ERA prior to deliberate release or being placed on the market. Recently, the European Food Safety Authority (EFSA) published two opinions on the relevance of the current EU ERA framework for GM plants obtained through novel genomic techniques (NGTs). Regarding GE plants, the opinions confirmed that the existing ERA framework is suitable in general and that the current ERA requirements need to be applied in a case specific manner. Since EFSA did not provide further guidance, this review addresses a couple of issues relevant for the case-specific assessment of GE plants. We discuss the suitability of general denominators of risk/safety and address characteristics of GE plants which require particular assessment approaches. We suggest integrating the following two sets of considerations into the ERA: considerations related to the traits developed by GE and considerations addressing the assessment of method-related unintended effects, e.g., due to off-target modifications. In conclusion, we recommend that further specific guidance for the ERA and monitoring should be developed to facilitate a focused assessment approach for GE plants.

## 1. Introduction

The ruling of the European Court of Justice (ECJ) in the case C-528/16 delivered in July 2018 clarified that plants developed by novel genomic techniques for directed mutagenesis are considered genetically modified organisms (GMOs) in the EU in accordance with Directive 2001/18/EC on the deliberate release and placing on the market of GMOs. The ruling also confirmed they are not exempt from regulations according to Article 3 in conjunction with Annex IB of the Directive (i.e., the “mutagenesis exemption”) [1]. In a broader sense, the decision established that organisms which are developed by methods of directed mutagenesis such as GE are subject to the current EU regulatory framework for biotechnology products. The EU biosafety framework was introduced in 1990 and underwent major amendments. In 2001 and 2003 the Directive 2001/18/EC and Regulation (EC) No. 1829/2003 on GM food and feed were introduced. In and 2013 and 2015 the Implementing Regulation (EU) No 513/2013 on requirements for the authorization of genetically modified (GM) food and feed and Directive 2015/412/EU providing EU Member States with the possibility to implement restricting measures on the cultivation of GMOs in their territories were adopted [2]. The decision of the ECJ was a major step in the long and heated debate in Europe concerning the regulation of organisms developed by novel genomic techniques such as genome editing (GE), but did not resolve all uncertainties regarding the regulation of such applications [3]. First, the ruling does not apply to all types of NGTs, which cover a diverse range of methods including cisgenesis, transgrafting and epigenetic engineering by methods of RNA-directed DNA methylation alongside GE [4]. Secondly, it was argued that the ruling does not resolve all pending questions regarding the practical implementation of the EU regulatory framework for GE organisms [5]. Subsequently to the ECJ ruling, the European Commission conducted a stakeholder survey in the framework of a study regarding NGTs including GE to address some of these issues. The recently published study, however, does not provide concrete policy recommendations for further discussion [6].

GE is mostly done through introducing DNA single- or double-strand breaks at specific loci of a target genome by a range of site-directed nucleases (SDN), with CRISPR-Cas-type nucleases being them most prominent among them [7]. Mutations are then introduced at these genomic sites by cellular DNA repair systems. The outcome of the genetic modification may be directed by template DNA sequences supplied in trans or by modifications of the used SDN [8]. SDN-based GE has quickly become a standard tool in molecular biology for a variety of uses, including fast–track plant breeding [9]. The discovery of the CRISPR-Cas system as a genome editing tool was awarded the 2020 Nobel Prize in Chemistry [10]. Due to its simplicity and accessibility, GE has been used at an increasing pace and scale for the development of genetically modified plants in recent years [11]. GE is believed to be of high importance for future plant breeding by certain stakeholders [12,13]. The regulatory uncertainties surrounding GE organisms, particularly the question of whether GE organisms are GMOs according to many existing biosafety frameworks, led to policy considerations and debates in most countries of the world and at the level of international organizations. The increasing use of GE in plant breeding at the global level made this debate more urgent [14,15,16,17]. Against the background of the different national systems for the regulation of GMOs, some countries, including a number of Latin American countries, have already introduced supplementary legislation to facilitate the determination of the regulatory status of individual GE applications with regard to the existing biosafety laws [18]. Some countries, such as Australia, have decided to exclude some types of GE applications from their regulatory framework for GMOs [19]. Other legislations such as the EU and New Zealand have sought decisions of their supreme courts to decide whether GE organisms are subject to their existing regulatory system for GMOs. In both cases, the court rulings have positively answered this question [15]. Canada is operating a regulatory system that is based on the novelty of the newly developed traits and the plausibility of hazards that may be associated with the use of modified plants as regulatory triggers. Canadian regulations for plants with novel traits accommodate GMOs as well as plants with novel traits established by GE or conventional breeding within the existing regulatory framework [14,15].

In all countries, the decision to regulate GE plants according to the existing GMO regulations is crucially relevant for the level of regulatory oversight for GE plants. These decisions are thus highly important for the particular risk assessment requirements applied for GE plants [15]. Thus, the current debate in the EU as well as in other countries focuses on two issues: the practical applicability of the current regulatory system for products of novel genomic techniques such as GE and the development of appropriate approaches for the assessment of food safety and the environmental risk assessment (ERA) of organisms developed by GE. This review is focusing on the latter question. Specifically, we discuss considerations regarding an appropriate risk assessment of the traits developed by GE approaches as well as any unintended effects of GE plants. We suggest that further specific guidance for the ERA and monitoring of GE plants should be developed. We note that considerations regarding the risk assessment for GE plants will inform the debate on options for further regulation [20].

## 2. Recent Considerations for the ERA of GE Plants at the EU Level

In their explanatory note addressing new techniques in agricultural biotechnology [7], the High Level Group of Scientific Advisors to the European Commission concluded that a highly diverse range of applications and possible products of GE and other novel genomic techniques need to be considered in the debate on regulatory approaches and the ERA. As a general conclusion, they suggested an appropriate ERA needs to address the following aspects in a case specific manner:Effects due to intended changes present in the modified plant;Effects due to unintended changes present in the modified plant;Effects due to the characteristics of the modified plant species and its interaction with the receiving environment;Effects due to the intended use of the modified plant.

Such considerations apply to the ERA, which is currently conducted for GMOs in accordance with Directive 2001/18/EC and related EU legislation, e.g., Regulation (EC) No 1829/2003 on GM food and feed. The scientific risk assessment for GMOs is based on guidance developed by the EFSA. Such guidance is available for (1) molecular characterization; (2) comparative assessment including agronomic, phenotypic and compositional characterization; (3) food and feed safety assessment; and (4) environmental risk assessment [13]. EFSA published a general guidance document for the ERA of GM plants in 2010 [21]. Furthermore, notifications for authorization of GM products in the EU need to conform to the information requirements as set forth in Implementing Regulation (EU) No 513/2013. This regulation implements elements of the existing guidance document on risk assessment of GM food and feed [22] in a legally binding form.

In 2012, EFSA delivered an opinion addressing the risk assessment of GE plants which contain site specific insertions of exogenous sequences (so called SDN-3 applications) [23]. Against the background of the ruling of the ECJ in case C-528/16, the European Commission tasked EFSA in 2019 with several mandates for opinions concerning emerging novel genomic techniques. A recently published review provides a brief overview on the mandates relevant to the risk assessment of organisms developed by GE [13]. In particular, two opinions and the related documents on the results of the consultation processes conducted for the respective draft opinions are pertinent to the discussion of an appropriate ERA approach for GE plants:The opinion on the applicability of the previous EFSA Opinion from 2012 on SDN-3 for the assessment of plants developed using SDN-1, SDN-2 and oligonucleotide directed mutagenesis (ODM), i.e., GE methods to typically generate small-sized random (SDN-1) or template directed (SDN-2) mutations at predefined genomic loci [24,25].The opinion on the evaluation of existing guidelines for their adequacy for the molecular characterization and ERA of genetically modified plants obtained through synthetic biology [26,27].

The second opinion discusses a wider range of applications than GE. It, however, addresses a low-gluten wheat plant produced by targeted mutations of multiple alpha-gliadin genes using CRISPR-Cas9 genome editing as one of the three case studies discussed in the opinion.

Due to limitations by the terms of reference, EFSA did not produce a new, stand-alone guidance document for the case-specific risk assessment of GE plants. Rather, the GMO panel stated whether the previous conclusions of the 2012 opinion on SDN-3 applications were applicable for any SDN-1, SDN-2 and ODM applications [13]. The 2020 opinion therefore recurred on the 2012 opinion on SDN-3 applications, which in turn recurred on the general guidance document on ERA. The opinions concluded that the general approach and the principles developed for the assessment of GMOs are relevant and applicable for GE plants. The opinions further stress the necessity of a case specific assessment and indicate that the existing assessment approaches need to be adapted with a view to the characteristics of the individual GE applications. However, no further (case-) specific guidance was provided in the EFSA opinions. This was noted by several comments during consultations, which called for further work to provide more detailed guidance [24]. The recent EFSA opinions also did not address the limitations and shortcomings of the existing assessment and monitoring approach for GMOs, which should also be considered for GE plants [28]. As highlighted in this review, some aspects of the current system may particularly affect the robustness of the assessment of GE plants: (1) the specific focus on newly expressed transgenic proteins, (2) difficulties concerning the choice of appropriate test organisms to assess any adverse effects of modified plants on the receiving environment(s) and (3) the testing of chronic effects, indirect effects and interaction effects focusing on individual new compounds rather than on the entire modified plant. The current limitations regarding the post-marketing environmental monitoring (PMEM) should also be considered with a view of GE plants [28].

## 3. Generic versus Case-Specific Considerations for the Assessment of GE Plants

The precautionary principle requires a case-by-case evaluation of the risks associated with GMOs. However, the current discussion concerning regulatory approaches regarding GE organisms and specifically regarding GE plants typically focuses on trying to establish classes to categorize GE organisms based on the GE technique used and the type of modification introduced [29]. Some countries used such an approach to specify GE applications which should be further regulated and thus be subject to a risk assessment according to the respective biosafety laws [14,18,19]. A crucial question, however, is whether general denominators of risk/safety are available which would allow for a conclusion on the safety of whole groups of applications instead of applying case-specific considerations for all individual GE applications.

In the following, we discuss some generic considerations with a view to their suitability for such classification, in particular:Considerations regarding the type of GE method (SDN-1; SDN-2/ODM; SDN-3);Considerations regarding the size of the introduced genetic changes;Considerations regarding the precision of the editing process;Considerations regarding the complexity of the introduced changes (i.e., the depth of intervention);Considerations regarding the novelty of the developed traits;Considerations regarding the speed of the development.

### 3.1. Considerations Regarding the Type of GE Application

The trigger to determine the status of their regulation in some countries, such as Argentina, Brazil, Chile, Colombia, Australia and more recently the USA, is based on considerations regarding the type of GE method which is used to create GE plants [14,15,18,19]. The underlying consideration is that only some GE applications-such as SDN-3 applications result in the integration of longer exogenous sequences into the genome of the modified plants [13]. Rostocks [13] argues that modifications introduced by SDN-1, SDN-2 and ODM in general would resemble mutations which may also be introduced by classical mutation breeding.

However, this does not take into account that the scale, scope and location of mutations which can be introduced by GE may differ quite significantly from those mutations which may arise spontaneously during conventional breeding. A recent review [30] shows that GE facilitates introduction of multiple mutations, such as the simultaneous editing of several genes/alleles (multiplex editing) or the editing of gene alleles that are inaccessible to conventional breeding.

Furthermore, the theoretical comparison of spontaneous mutations with modifications introduced by GE does not consider the specific hazards that may be associated with a particular mutational change. The occurrence of hazards thus would not be correlated in all cases with an exogenous origin of the introduced DNA sequences.


**Thus, case-specific considerations seem to be more appropriate than considerations based solely on the type of GE applications.**


### 3.2. Considerations Regarding the Size of the Introduced Genetic Changes

Another general consideration regarding genetic modifications via GE is that the mutations introduced by SDN-1, SDN-2 and ODM applications typically are of small size. In some cases, only minimal sequence changes called single nucleotide variants (SNVs) are introduced [31]. As discussed in Section 3.1, some regulatory frameworks exempt such GE applications, in particular SDN-1 applications, from the scope of their biosafety regulations.

The respective regulations, however, are not consistent upon comparison. In the USA, only GE plants with SNVs are exempt from oversight by USDA APHIS. Organisms with two or more base pair changes do not qualify for automatic exemption [14]. Decision criteria for regulatory exclusion of individual modified organisms from the biosafety laws in Argentina, Brazil, Chile and Colombia exclude small sized SDN-1, SDN-2 and ODM modifications (SNVs and small insertions/deletions). Australia only excludes SDN-1 applications, while regulating all GE applications using repair template sequences, including SDN-2 and ODM applications. A major reason for introducing the specific regulations, particularly in Australia, was to create a simple and enforceable system for determination of the regulatory status of individual GE applications by developers and/or authorities [15,19].

However, **the size of the modification cannot be regarded as a reliable denominator of risk/safety of the specific modifications present in individual GE plants**. On the contrary, it is well known that even small DNA sequence changes can significantly impact the function and effects of the modified genes within the context of the GE plant. Thus, the High Level Group of Science Advisors concluded that the risk associated with particular sequence changes can only be assessed case-by-case [7,32].

Considerations regarding the size of the sequence modifications introduced by GE are more relevant to the question of whether the respective GE plants can be identified as a specific product, i.e., unanimously distinguished from other plant varieties by state-of-the-art detection methods [33]. However, GE plants with multiple small modifications or larger modifications may be identified by such methods [34]. The possibility to analytically identify a specific GE product is less relevant for the ERA than the ability to determine the environmental exposure to certain GE plants during PMEM and other enforcement requirements.

### 3.3. Considerations Regarding the Precision of the Editing Process

GE methods promise to introduce genetic modifications at specific genomic locations with a much higher precision than other methods for mutagenesis [13]. However, **the specificity of the used GE systems is not absolute. All GE methods are known to have the potential to also introduce off-target modifications [11,20,35]**. As acknowledged by Rostoks [13], some off-target activity must be expected with GE. He also indicates that the methods to predict such activity in silico are not absolutely reliable. Furthermore, integration of extraneous DNA elements at DNA-breakpoints such as off-target cleavage sites may occur [13].

The precision of GE, i.e., the specificity for GE to happen only at intended target sites, is therefore a relevant denominator for the potential occurrence of unintended modifications. Such modifications might be associated with adverse effects, thus the identification and characterization of off-target modifications in the final plant product is relevant for the assessment of unintended effects [36,37].

### 3.4. Considerations Regarding the Complexity of the Introduced Changes

GE plants described in the scientific literature contain a range of different modifications to address different breeding objectives [11]. Only some GE plants contain single or few modifications that result in single, specific phenotypic changes. A significant number of GE plants were modified to facilitate complex physiological or phenotypic changes [20]. In particular, the modification of genes, which facilitate multiple different (pleiotropic) effects or which target genes involved in regulatory responses in the parental plants, may give rise to complex phenotypical changes that may be challenging to identify or assess. Another category of GE applications, which facilitate a higher depth of intervention, are multiplexed GE applications to create complex physiological, developmental or morphological changes. A number of such applications were described in recent reviews [13,20]. Examples include GE wheat with modifications in six homeoalleles of a gene (*TaMLO*) to increase resistance against powdery mildew, GE wheat edited in multiple alpha-gliadin genes resulting in a low gluten content and GE wild tomato modified in several genes for de novo domestication. The latter is a novel approach for the rapid development of tomato varieties that combine desired traits found in wild tomato plants, such as resistance toward pathogens or salt tolerance, with agriculturally favorable traits occurring in domesticated tomato varieties [38].

**A high depth of intervention and/or complexity of the introduced changes may serve as an unspecific general indicator that a robust, comprehensive ERA is required**. With regard to a respective case study (a low-gluten GE wheat), EFSA concluded that such applications go far beyond any GM plants previously assessed. However, EFSA also concluded that the existing requirements according to the current ERA approach for GMOs are adequate and sufficient for such types of GE plants [27].

### 3.5. Considerations Regarding the Novelty of the Developed Traits

A wide range of different traits have been developed by GE in different plant species. Some of these traits are related to traits already occurring in crops produced by conventional breeding or in wild relatives which could be crossbred. Other traits described in the scientific literature are similar to ones established in GM plants, e.g., herbicide or disease resistance [20]. However, a significant number of traits were not previously established by conventional breeding or other biotechnological methods, such as classic GM technology or the silencing of endogenous genes through RNAi methods. The latter category thus contains plants with novel traits. Less knowledge is usually available for plants with novel and untried traits than for GE plants that are comparable to conventionally bred plants or already assessed GM plants [20]. In particular, knowledge from practical experience in agricultural production, from observation of related wild plants and/or from previous risk assessments may be lacking. The Plants with Novel Trait (PNT) regulation implemented by Canada mandates a case-specific risk assessment of PNTs.

The available level of knowledge and/or history of safe use needs to be considered for the assessment of the intended modifications and the resulting intended traits with comparable traits in similar or related crop or plant species. However, such information does not relate to any unintended effects due to the modification process by GE. **Familiarity thus cannot serve as a general denominator of overall safety**. Novelty of the trait indicates the need for new data to assess risk issues relevant for the particular GE plant.

### 3.6. Considerations Regarding the Speed of the Development

GE is expected to reduce development time considerably, particularly for plants harboring multiple independent modifications [39]. It is estimated that the development time for GE plants is substantially shortened compared with classical GM plants and conventionally bred plants. Development time for GE plants is estimated to be 4–6 years in comparison with 8–12 years for GM or conventional plants [9]. **When fewer backcross generations are necessary to develop elite varieties from GE plants, the possibility that unintended modifications are removed during subsequent crossbreeding steps is decreased**. This is particularly important for applications facilitating the direct editing of elite lines, the editing of agricultural plants that are predominantly propagated vegetatively and for GE perennial plants with long generation times such as trees [20]. The speed of the development process may be considered an unspecific and indirect indicator of risk/safety, since a shorter development time of GE plants is constraining the time for the assessment of any unintended and unexpected effects.

### 3.7. Conclusions Regarding the Appropriate Approach for Risk Assessment

Based on the discussion of the suitability of general denominators for risk/safety, we argue that **a case-specific risk assessment within the current regulatory frameworks for GMOs should be conducted. This is considered a better option than to exclude certain classes of GE application from GMO regulation and from the established systems for risk assessment under these regulations**.

However, some generic considerations can provide relevant input on how to focus the ERA on relevant risk areas (assessment of intended traits or assessment of unintended effects) and on specific risk issues according to the existing guidance [21].

## 4. Considerations for the Case Specific Assessment of GE Plants

When designing a case-specific assessment, the following characteristics of GE applications need to be considered:The different GE techniques (SDN-techniques, ODM) and the various approaches for application that are available or in development and used to modify plant species (SDN-1, SDN-2, SDN-3, base editing, prime editing, epigenetic engineering). An overview on these approaches, their different characteristics as well as recent developments is given, e.g., by Adli [40], Anzalone and coworkers [41] and in a recent study by the Joint Research Centers of the European Commission (JRC) [29].The specific characteristics of such GE approaches with regard to their target specificity [20] and their ability to modify genomic locations that are not accessible to conventional breeding [30].The wide range of plant species that can be modified by GE approaches. This range includes a multitude of plants used in agriculture and forestry, as well as a range of non-crop plants [20,42].The broad range of traits that is under development [9,11]. Some of these GE plants are already marketed in certain countries or may be placed on the market in the near future [12]. A number of these traits are novel, some are highly complex.The interactions of the individual GE plants with the respective receiving environments, taking into account the specific conditions of their use and the possibilities for unintended introduction into non-managed habitats.

In addition to the principles applied to the ERA of GMOs according to Directive 2001/18/EC, the ERA conducted for GE plants should also be based on the characteristics presented above. For the design of a case-specific ERA, two sets of considerations need to be taken into account, regarding GE plant x environment interactions:Trait-related considerations to assess the effects of the intended trait(s).Method-related considerations to assess the unintended effects.

### 4.1. Trait-Related Considerations

The level of risk associated with a GE plant depends significantly on the effects of the developed trait(s) on the overall characteristics of the modified plant species [32]. Thus, a case-specific ERA must specifically consider the introduced trait(s) as well as the plant species that are modified. Recently, some systematic reviews of the scientific literature on GE plants have been published [9,11,20]. These reviews indicate that:**A wide range of plant species is used for GE**, either as model organisms for scientific research and method development, such as *Arabidopsis thaliana*, tobacco and rice, or plants that might be used as ornamental plants or in agriculture, forestry and industrial production. Examples for the latter groups include apple, barley, camelina, cassava, cotton, cucumber, flax, grapefruit, grapevine, legumes (soybean and barrelclover), maize, oilseed rape, opium poppy, poplar, potato, rice, rubber dandelion, red sage, tobacco, tomato, watermelon and wheat (bread wheat and durum wheat) [20].The GE applications in such plants are at different stages of development. Most reports in the literature are accounts of early development including proof of concept studies [20]. **A rising number of GE plants are currently developed for marketing. However, only a few are commercialized [12,43]**. The latter groups are particularly interesting for regulators to keep track of since they may be presented for regulatory assessment in the near future.**A broad range of traits are considered for development**. The systematic review conducted by Modrzejewski and coworkers [11] listed 101 GE applications that might be relevant for the use in agriculture in the near future. A recent analysis of these applications indicates the following [44]. One major focus is on the development of traits that increase the agronomic value of crop plants (increased yield, improved storage quality, enhanced crop development; 38% of applications), or alter the composition of the plants (e.g., reduced lignin content, altered fatty acid composition; 28% of the applications). Sixteen percent of applications concern different approaches to increase the resistance to biotic stress (particularly for resistance to fungal or bacterial pathogens) and 8% are for modified content for industrial purposes (improved starch quality, altered oil composition). Another 8% are for plants with resistance to broadband herbicides (e.g., herbicides containing glyphosate or ALS-inhibitors) and 5% of the applications are for enhanced abiotic stress tolerance (e.g., tolerance to drought or salt stress).**Most of the applications are aimed at knocking out the expression of plant genes** involved in the above-mentioned processes via SDN-1. Fewer applications are for functional modification of genes (SDN-1, SDN-2, ODM applications). Other applications of agronomic importance, e.g., applications to develop herbicide resistant plants, are based on SDN-3 approaches [20].**A majority of the current developments are applications to modify a single target gene**, or all alleles of such genes present in the target plant. However, there is a significant and increasing number of applications for multiplexing. Examples for such developments are provided, e.g., in Kawall et al. [44] and Eckerstorfer et al. [20]. Applications with a higher depth of intervention are developed for different purposes, including altered composition, increased yield and developmental and morphological alterations beneficial for agricultural use.**A significant number of the traits developed by GE need to be considered novel**. Some of these developments are not feasible by conventional breeding approaches [30].

We conclude that the assessment of some of these traits will be challenging. EFSA came to the same conclusion in their case study of a complex modified low-gluten wheat plant modified in multiple genes by a CRISPR/Cas-based method [27]. Such applications differ significantly from any plants which were assessed previously and would require a comprehensive approach for risk assessment including ERA and food/feed safety assessment. Similar conclusions are drawn in a recently published study on a GE *Camelina* plant with altered fat composition [45]. A focused but robust ERA also needs to be provided for GE plants with traits that enhance their biological fitness or alter their reproductive properties. In addition, applications which provide a fast-track development of crops from non-domesticated wild forms [38] should be considered novel crops and should undergo a comprehensive risk assessment [20]. In contrast to other GE applications, no history of safe use and possibly only limited scientific data will be available for most of the above-mentioned complex GE applications.

Existing experience (“familiarity”) with similar plant x trait combinations should be considered when a similar use of the corresponding GE plant is intended. In some cases, familiarity may be available with a specific trait, which was already used in conventionally bred crops employed in agricultural production for some time, particularly with respect to food and feed safety. The availability of a history of safe use regarding environmental effects is less likely, considering the complex nature of plant x trait x environment interactions. In some cases, however, such as GE herbicide resistant plants, conventional counterparts exist and the respective experiences with the environmental effects of such conventional herbicide resistant plants and its management should be considered in the ERA [46].

However, it needs to be emphasized that the concept of familiarity should be used as a tool to strengthen the case-specific approach to risk assessment. Like EFSA concluded for the concept of substantial equivalence [21], it may be used as a starting point to determine risk assessment needs and the requirement for newly established data rather than as an endpoint of the assessment.

### 4.2. Method-Related Considerations

Method-related considerations should be applied to facilitate the assessment of unintended effects. As suggested previously, the overall process of modification should be considered, including the steps to introduce GE tools for modification into the target plant cells or tissues. Duensing and coworkers [32] indicate that, in most cases, transgenic constructs are introduced into plant cells transiently or integrated into the genome of the recipient cells to express the required GE tools. While such integrated constructs are typically removed in breeding steps subsequent to GE, **the absence of exogenous constructs or secondary modifications (spurious insertions) need to be confirmed [44,47]**. Lema [47] recommends that routine approaches using Southern hybridization methods should be used to assess spurious insertions. Alternatively, the absence of exogenous sequences can be assessed by whole genome sequencing (WGS) data and bioinformatics analysis [47].

One well known aspect of GE applications is that the used nucleases do not recognize the targeted genome loci with absolute precision, resulting in some level of off-target activity [35,36,37]. A recent report from the JRC provides a detailed discussion of the available knowledge regarding off-target activities associated with the different GE methods and tools [29]. The report highlights that **the presence of off-target modifications has not been well-studied for a number of GE applications**, in particular for GE applications of recently developed methods or for methods which are only used in a limited number of applications. Thus, the general notion that GE methods are inducing off-target modifications with a low probability is based on a limited amount of reported data [29]. The JRC report also highlights that off-target activity is not only found with SDN introducing DNA double strand breaks, but essentially with all existing GE methods. For example, recent publications address the off-target activity of base-editing enzymes [48] and SDNs that are modified for epigenetic engineering [49].

EFSA also discussed off-target-activity of SDNs [13,25]. As an overall conclusion, they considered the level of off-target activity lower than the mutation rate due to classical mutagenesis [13]. Furthermore, they referred to the availability of strategies to increase the precision of editing and to remove off-target modifications in subsequent crossbreeding steps [13]. Based on such general considerations, EFSA considered the overall risk low and recommended no detailed risk assessment approach in their recently published opinion [25].

However, not all GE approaches can be designed to minimize the occurrence of off-target modifications. In cases aimed at simultaneously modifying a number of genomic loci with slightly different target sequences in a quite simple way, a high level of specificity for all single targets is not feasible. In such applications, genome editing tools with a lower level of specificity (i.e., precision) are employed, which recognize all the slightly different target sequences with appropriate efficiency. Such intentionally “dirty” approaches are a straightforward approach to simultaneously modify different genomic targets sites, which are not perfectly homologous, e.g., different members of a gene family [45]. DNA breaks introduced by off-target activity may facilitate the insertion of extraneous DNA sequences, which in turn may lead to unintended effects [44,47].

Additionally, the introduced off-target modifications may not be readily removed in all cases. This is particularly true for approaches which require fewer backcrossing steps than conventional breeding schemes or are designed to avoid such backcrossing steps altogether, i.e., approaches for “quick” breeding schemes. Also, secondary modifications introduced by GE systems in the vicinity of the intended genomic target site should be appropriately assessed. Such modifications are tightly linked to the intended traits and are not easily lost during subsequent breeding [20,44,47]. Therefore, **we suggest drafting further guidance for the assessment of unintended effects of GE modifications**.

A number of methods, including WGS, are available for a targeted and untargeted analysis of unintended modifications and should be considered for a case-by-case evaluation [20]. Specifically, such tools should be applied if the characteristics of the used GE approach suggest a higher probability for off-target modifications to occur in a GE plant. In particular, “quick and dirty” GE approaches, i.e., GE approaches with a higher level of off-target activity and fewer subsequent breeding steps to remove secondary modifications, should be thoroughly assessed for unintended off-target modifications and associated adverse effects. The previously proposed 10 step approach to assess unintended effects described by Eckerstorfer and coworkers [20] is considered a good starting point for a focused assessment of unintended effects. In addition, recommendations by the French Haut Conseil des Biotechnologies [50], Kawall and coworkers [44] as well as by Lema [47] concerning the assessment of off-target modifications and spurious insertions should be considered to develop appropriate guidance.

## 5. Implications for Regulatory Approaches for GE Plants

As outlined in the above chapters, the emerging GE applications present a number of challenges for regulators, risk assessors and policy makers. For the policy makers and regulators, one challenge is to ensure a legislation based on regulatory triggers that are simple to use, unambiguous and easily enforceable, yet flexible enough to cope with emerging techniques such as GE. From a risk assessment point of view, the main challenge is that such a legislation should ensure an adequate assessment of the diverse combinations of plants and traits obtained by GE. Such an approach must take into account the associated risks in accordance with the objectives of established biosafety legislation, i.e., a high level of protection of human and animal health and the environment. However, the regulatory solutions developed by policy makers do not necessarily resolve the challenges regarding the ERA and environmental monitoring of biotechnology applications. As discussed in the previous sections, the risks associated with GE plants is correlated with the newly developed traits and/or unintended effects resulting from characteristics of the specific GE approach used to establish a particular GE plant rather than with a certain type of GE approach per se (e.g., SDN-1, SDN-2/ODM, SDN-3). Certain approaches that are complex and/or fast and/or dirty may be associated with a higher risk.

This results in challenges to develop a regulatory framework that is broad enough to include all types of GE applications, while providing enough flexibility to focus the attention and resources of risk assessors on the characteristics that are particularly relevant for the GE plant in question.

**In general, we consider the principles and the case-specific approach provided by the current framework for GMOs to be appropriate for the emerging GE applications**. This is in accordance with the recent EFSA opinions [25,27].

However, **we consider the exclusion of whole classes of GE applications from the existing regulatory frameworks for biotechnology applications, e.g., the current EU framework for GMO regulation, a poor option from a biosafety perspective**. As discussed previously [15], other applicable regulation is insufficient to address biosafety issues. In fact, the European seed legislation, food and feed law as well as the plant protection law and plant variety protection law are neither individually nor collectively able to ensure an assessment and control of possible negative environmental impacts of NGTs [51]. Such regulations that apply to all agricultural plants, genome edited or not, are thus not well suited to provide an appropriate framework for case-specific risk assessment according to the high safety standards implemented in the respective GMO legislation.

## 6. Conclusions

Our review indicates the challenges faced by policy makers, regulators and risk assessors to provide an appropriate framework for the risk assessment of GE plants. The risk associated with individual GE applications will be highly variable. While the effects of some GE applications may be well known from conventional varieties with similar traits, other GE applications could be associated with plausible risk issues and may be more challenging to assess and monitor. The latter group will likely be comprised of GE plants with complex and novel modifications as indicated by EFSA [27] and other authors [20,44].

Considering the wide ranges of plant species and the GE methods and traits that need to be considered, there is no safety by default for whole groups of GE applications encompassing different individual GE organisms. Biosafety considerations should instead be based on an appropriate ERA prior to the release of GE plants into the environment.

The case-specific approach incorporated in the EU regulatory framework is a viable way forward provided that further guidance for the risk assessment of GE applications is developed. The existing guidance developed by EFSA and their initial work on GE applications is not sufficient to address these challenges, but rather a starting point for further efforts. In this review, **we argue that general considerations concerning risk/safety of all GE applications or of different classes of GE applications are insufficient to address the challenges at hands. Instead, we suggest that a focused case-specific approach is followed to provide a robust risk assessment of individual GE plants**. This ERA approach should focus on risks that may plausibly manifest themselves in the phenotype or the interaction with the environment of a particular GE plant. To this end, we suggest that two sets of considerations are considered: (1) trait related-considerations to assess the effects associated with the newly developed trait(s); and (2) method-related considerations to assess unintended changes associated with the intended trait(s) or with other modifications in the GE plant. Important aspects concerning both sets of considerations are outlined in Box 1.

**Based on these considerations, further guidance should be developed to ensure the high safety standards provided by the current regulatory framework for GMOs in the EU for GE plants in an adequate and efficient way, taking into account the existing knowledge and experience in a case-specific manner**. This guidance should thus strengthen the case-specific approach that is recommended by numerous EU and Member States institutions. The precautionary approach of the existing EU GMO regulations should not be weakened by excluding whole groups of GE applications from their scope without having regard to the characteristics of the individual GE plants.

Box 1: Crucial aspects for a two-pronged assessment strategy to address trait-related effects and method-related modifications, respectively.
(1)The assessment of effects associated with the newly developed trait(s) in GE plants should consider, among others:The level of knowledge and familiarity with the particular crop and trait combination needs to be considered. As indicated in Section 4.1, only limited scientific knowledge is available for some GE applications.Some applications may lead to changes in agricultural management; possible indirect effects resulting from their use need to be addressed during the ERA.Complex GE modifications should be thoroughly scrutinized regarding adverse environmental effects resulting from these changes. A robust assessment should be provided for physiological effects of multiple simultaneous changes (multiplexed GE) and for regulatory effects of the introduced modifications on morphology, development and reproduction of the GE plant.The ERA conducted for GE plants should also address secondary effects associated with the intended trait(s). This should encompass pleiotropic effects of the intended trait(s).(2)The assessment of method-related unintended changes associated with the intended trait(s) or with other modifications in the GE plant should take into account the following aspects:The available body of evidence with regard to off-target-effects, their occurrence and their identification as indicated in Section 4.2.The likelihood that off-target modifications are still present in the final breeding product. This likelihood may be higher with fast-tracked breeding applications, i.e., aimed at modification of elite lines, modification of vegetatively propagated crops, and modification of plant species with longer generation cycles such as trees.The available information on unintended secondary modifications introduced by GE systems in the vicinity of the intended genomic target site. Such modifications are tightly linked to the intended traits and are not easily lost during subsequent breeding steps.The available recommendations on how an assessment of unintended and off-target effects may be conducted and which kind of aspects should be considered in the framework of the assessment.

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
