# Peer review of "Biosafety of Genome Editing Applications in Plant Breeding: Considerations for a Focused Case-Specific Risk Assessment in the EU"

_biotech, 2021, doi:10.3390/biotech10030010_

Round 1

Reviewer 1 Report

The review is well written and comprehensive, with relevant sources and literature taken into account.  Minor spell check is required throughout the text, e.g. In the title of the article “Eu” should be corrected into “EU”.  Otherwise, I recommend accepting the article in its present form.

Author Response

The provided review is kindly acknowledged.

The title is corrected as indicated. An editorial check was conducted throughout the whole manuscript to correct spelling errors and enhance the readability of the text.

The text of the conclusions was shortened and the previously included specific recommendations as regards trait-related effects and method-related changes are now presented in a Textbox (Box1) complementing the section.

Reviewer 2 Report

The title should be "Biosafety of Genome Editing Applications in Plant Breeding-Considerations for A Focused Case-Specific Risk Assessment in The EU".

In line 370, "legumes(Medicago truncatula)" should be "legumes(Medicago truncatula and Glycine max). Accordingly in the line 372, "soybean" should be deleted.

 I found some spelling mistakes. For example, in line 536, "und" should be "and". Please check the whole manuscript.

On the other hand, the section of conclusion is too long.

In general,  the readability of the manuscript is not good.

Author Response

The provided review is kindly acknowledged.

The title is corrected as indicated.

An editorial check was conducted throughout the whole manuscript to correct spelling errors, including the examples indicated by the reviewer (line 536).

The changes suggested by the reviewer regarding line 370/372 were addressed – thanks for your suggestions! For consistency and brevity only common plant names are listed.

The whole text was scutinized to enhance the readability of the text. E.g. longer sentences were split and the use of brackets was avoided.

The text of the conclusions was shortened and the previously included specific recommendations as regards trait-related effects and method-related changes are now presented in a Textbox complementing the section.

Reviewer 3 Report

The authors are competent employers of the federal organizations in the field of environmental protection and biosafety from five European countries. The paper provides information on the existing regulatory framework for using Genome Edited (GE) organisms. The authors note uncertainties regarding the regulation of such applications and discuss the possible risks of introducing GE plants into global circulation. They note the need for further research and discussion in this area and the development of further guidance on the use of GE technologies. The article will certainly be useful to a wide range of biologists and specialists in the field of environmental protection and biosafety.

Author Response

The provided review is kindly acknowledged.

An editorial check was conducted throughout the whole manuscript to correct spelling errors and enhance the readability of the text.

The text of the conclusions was shortened and the previously included specific recommendations as regards trait-related effects and method-related changes are now presented in a Textbox complementing the section.

Round 2

Reviewer 2 Report

I noticed that you have revised your manuscript. Presently the readability of your manuscript has been improved.